# The importance of human factors in therapeutic dietary errors of a hospital: A mixed-methods study

**Amanullah Khan[1], Sidra Malik[2], Fayaz Ahmad[1], Naveed Sadiq** 📀 [1] *

**1** Institute of Public Health & Social Sciences, Khyber Medical University, Peshawar, Pakistan, **2** Riphah International University, Islamabad, Pakistan

* naveedsadiq@gmail.com

**Data Availability Statement:** The de-identified data for this manuscript could be accessed at the following DOI link, publicly available: https://doi.org/10.7910/DVN/2HGGOS.

## Abstract

An accurate therapeutic diet can help people improve their medical condition. Any discrepancy in this regard could jeopardize a patient's clinical condition. This study was aimed to determine prevalence of dietary errors among in-patients at an international private hospital's food department, and to explore causes of error to suggest strategies to reduce such errors in the future. Thus, a sequential explanatory mixed-methods study was carried out. For the quantitative part, secondary data were collected on a daily basis over one-month. For qualitative data, errors arising during the meal flow process were traced to the source on the same day of error followed by qualitative interviews with person responsible. Quantitative data were analyzed in SPSS v.25 as percentages. Qualitative data were analyzed by deductive-inductive thematic analysis. Out of a total of 7041 diets, we found that only 17 had errors. Of these, almost two-thirds were critical. Majority of these errors took place during diet card preparation (52.94%), by dietitians (70.59%), during weekdays (82.35%), breakfasts (47.06%), and in the cardiac care ward (47.06%). The causes identified through interviews were lack of backup or accessory food staff, and employee's personal and domestic issues. It was concluded that even though the prevalence of dietary errors was low in this study, critical errors formed majority of these errors. Adopting organizational behavior strategies in the hospital may not only reduce dietary errors, but improve patients' well-being, and employee satisfaction in a long run.

## Introduction

The World Health Organization (WHO) defines patient safety as "a health care discipline that emerged with the evolving complexity in health care systems and the resulting rise of patient harm in health care facilities" [1]. The goal of patient safety is to prevent and minimize risks, errors, and harm to patients while providing health care [1]. Some of the examples to threat of patient safety include, but not limited to, medication errors, health-care associated infections, unsafe surgical procedures, diagnostic errors, radiation errors, sepsis, etc. [1]. Healthcare organizations strive to deliver quality and safer healthcare to patients to ensure optimum treatment

**Funding:** The authors received no specific funding for this work.

**Competing interests:** The authors have declared that no competing interests exist.

outcome which reflects on the quality of the hospital and its staff [2]. This is why patient safety is one of the important components identified by Institute of Medicine (IOM) and WHO that outline principles for the healthcare organizations to provide quality healthcare to the patients [3], thus minimizing preventable harm and reducing risks during the delivery of healthcare [4,5]. A complexity health-care system makes humans more vulnerable to errors [1]. It is estimated that 10% of patients in high income countries face preventable adverse events during their hospital stay [5]. In low-middle income countries, the adverse health events during hospital stay result in 2.6 million deaths annually [5]. Dietary errors are one of the main preventable risks to patient safety [6–8]. Dietary errors are defined as meals containing one or more therapeutic dietary items other than the recommended one [6]. According to a study in Australia, 8% dietary errors were recorded in the hospital's food delivery system [6]. Another similar study reported that about 20% of the therapeutic meals served to in-patients in a metropolitan tertiary hospital, were inaccurate and 64.8% of these errors were critical in nature that could claim the life of a patient [8].

A therapeutic diet is one of the essential components of latest clinical treatment [9]. Patients are prescribed these diets to cope with and/or recover their health while being admitted in hospital [10], [11]. Therefore, therapeutic nutrition plays a key role in recovery of patients [12]. On the other hand, provision and consumption of inaccurate therapeutic diet could interfere with the patients' treatment and may pose a threat to their wellbeing [10,11,13]. For example, a patient who is advised a liquid diet, receives and consumes a semi-solid diet. There are critical and non-critical diet errors depending on the severity of the side effects it inflicts on patients' health. A patient receiving a food item which a patient is allergic to is a critical diet error. Critical diet errors can result in severe deterioration of a patient's health and may even cause death. Another instance could be that consumption of a thick texture food by a patient on liquid diet could end result in aspiration or choking, or consumption of high fiber diet by a patient who is on low fiber diet could exacerbate the gastrointestinal disease. Allergic reactions could also be a serious adverse effect of consuming incorrect therapeutic diet. Therefore, it is imperative that these errors do not go unchecked.

The computerized system has shown to minimize errors in the hospitals' meal delivery process [6]. However, errors are inevitable even in the computerized systems as these systems are operated by humans and to err is human. This study reports an account on the accuracy of meals delivered to in-patients, needing a therapeutic diet, admitted in one of the top international for-profit hospitals with strictly implemented American hospital standards. We aim to assess the prevalence of errors in the provision of therapeutic diets to in-patients, and to explore the underlying causes to suggest strategies to reduce the chances of such errors. We expect that this study will serve as a guide to enhance patient safety, improve patient satisfaction, and effectiveness of the meal delivery systems in international level hospitals located in low-middle income countries.

## Methods

We employed a sequential explanatory mixed-methods study design from 1st March 2019 to 31st March 2019. This design entails a quantitative approach followed by qualitative inquiry to explain the quantitative findings and to enhance the utility of findings for the concerned quarters [14]. We collected quantitative data by observing all meals that were provided to the inpatients for one month. For qualitative data, we interviewed the sources of errors- the respective staff in the food department of the hospital. The study tool used for this study could be accessed in the supporting document 1 titled "Study Tool". The study was conducted in a private hospital, located in the capital of Pakistan.

**Table 1. Classification of dietary errors as per hospital's criteria before the consumption of meals.**

| Advised Diet | Received Diet | Error Type |
|---|---|---|
| Regular | Soft/Full Liquid/ Semi Solid/Clear Liquid | Non-Critical Error |
| Soft | Full Liquid/ Semi Solid/Clear Liquid | Non-Critical Error |
| Semi Solid | Full Liquid/Clear Liquid | Non-Critical Error |
| Full Liquid | Clear Liquid | Non-Critical Error |
| Semi Solid | Regular/ Soft | Critical Error |
| Full Liquid | Regular/ Soft/ Semi Solid | Critical Error |
| Clear Liquid | Regular/Soft/Full Liquid/ Semi Solid | Critical Error |
| Allergic Patient | Received Allergens | Critical Error |
| Patient on intolerance Diet | Received any item that cause indigestion | Critical Error |
| NPO (Nil per Oral) | Received food item | Critical Error |
| Neutropenic Diet patient | Received raw/ uncooked item | Critical Error |

The food preparation and its delivery to the patients involved different sections of the food department in the hospital. The relevant clinical staff recommended the diet of the inpatients in their clinical notes, based on patient's need. The respective unit representative of the ward entered the diet in the computer system with patient details. The computer system then generated requests to the hospital food department for food preparation as per provided requirements.

There were two ways of tracking the errors in the system (Tables 1 & 2). When an error was noticed before consumption of the inaccurate meal (Table 1), the source of error was tracked and identified and the inaccurate diet was replaced with the correct one. If the error was noted after the consumption of inaccurate meal as a result of patient developing any adverse symptom, then it was checked with biochemical analysis to determine the dietary error. The list of biochemical analysis against all the advised diets are given in Table 2.

There were a total of five check points in the system and dietary errors could occur at any of these five points:

1. Diet entry:
   Diet entry was the first point where patient's diet was entered into the system by the Unit Representative (UR) and was the primary source of information provided to the dietary staff.

2. Diet card preparation:
   A hard copy of detailed diet information was then forwarded to the dietitian who made diet cards based on the information provided.

3. Meal packing & tagging:
   The kitchen received requests for food through the diet card. The kitchen personnel packed

**Table 2. Classification of dietary errors as per biochemical reports after wrong meal consumption according to hospital's criteria.**

| Recommended Patient Diet | TEST | Critical Values | Error |
|---|---|---|---|
| **Diet Recommended by the American Diabetes Association** | Blood Glucose Level | 380mg/dl (Type II), 500mg/dl (Type I) | Critical |
| **Low salt** | Blood Pressure | 180/110mmHg | Non-Critical |
| **No salt** | Blood Pressure | 210/110mmHg | Critical |
| **Low Cholesterol** | Lipid Profile | More than 210mg/dl | Non-Critical |
| **No Cholesterol** | Lipid Profile | >240mg/dl | Critical |
| **Renal** | Chem 7 | 6.5mEq/L | Critical |

and tagged food according to the diet card, for example, renal diet, no salt or no chili diet, etc.

4. Tray preparation:
   The tray was prepared according to the diet card and cross checked by the dietician who tallied the diet card with tag on the meal placed in the tray ready to be served to patient.

5. Meal provision:
   The checked diet was then carried out in a heated trolley to thepatient room. Before serving it to the patient, it was again cross checked by the floor staff with patient notes. If an error was detected, it was immediately reported to the food department for replacement. In case the error went unnoticed and the patient consumed the diet, the later vital statistics of the patient might indicate the incorrect diet.

## Quantitative study

Census sampling technique was used for the collection of quantitative data over a period of one month (March 2019). The dietary record was collected on a daily basis for all the three meals per day per patient. Any dietary error arising on a particular day was marked for subsequent qualitative interview on the same day. The dietary information of patients admitted for at least 24 hours to any of the seven wards (medical, cardiac care unit, kidney transplant, gastroenterology, orthopedics, surgical, and neurology) of the hospital during the study period was included in the study. The dietary information from the comatose patients, and patients on nasogastric feed was excluded from the study. The reason being that these diets used to come from the hospital's pharmacy and not the hospital's kitchen, e.g. glucerna, and such diets were not a part of the hospital's meal flow process.

We collected quantitative data by observing all meals provided to inpatients on a structured proforma, designed based on the variables that were either available and/or could be collected from the hospital system. The outcome variable was "diet error", a binary variable with two possible outcomes: Critical / Non-Critical Errors. The definitions for critical/non-critical errors were based on the definitions provided by the hospital (Tables 1 & 2).

There were a total of five independent variables in this study, all categorical in nature. The accuracy of all the therapeutic meals were assessed at five main points in the food delivery system namely "point of error": (1) Diet entry; (2) Diet card making; (3) Meal packing; (4) Tray preparation; and (5) Meal provision. The "responsible staff" included the unit representative, dietitian, cook, and service aide. The "Day of the Week" was a binary variable, defined as 'Weekday' and 'Weekend'. The "Meal Type" included breakfast, lunch, and dinner. The "Ward" variable was composed of five levels: renal, cardiac, gastroenterology, orthopedics, and surgical wards.

All the quantitative variables were categorical in nature. The row percentages were calculated for all the independent variables against the outcome variable. We calculated frequencies and percentages for all the critical and non-critical errors using SPSS version 25 [15]. The Fisher's exact test based p-values were also reported to find out any association between the independent variables and the outcome variable.

## Qualitative study

The qualitative data were collected through in-depth interviews (IDIs). We employed purposive sampling technique following the concept of saturation. First, we detected the point of error and then the person responsible for error was approached. We explained the purpose of

the interview/research and subsequently obtained oral consent for the interview to seek the potential causes and reasons of the dietary errors. For the face-to-face IDIs, a semi-structured interview guide was developed, based on the consensus of the authors and feedback from a qualitative research expert (apart from the one who's leading the qualitative part of the study). A total of three questions were asked from all the participants, with little differences in probes depending upon the point of error. Each interview lasted for up to 30 minutes. The IDIs were conducted in the local language by two Masters' students trained in qualitative research. Interviews were audio recorded, transcribed verbatim, and translated into English language. The Data were analyzed using a combined deductive-inductive thematic analysis approach [16]. The IDIs were transcribed by two researchers under the supervision of a senior qualitative researcher followed by debriefing sessions including cross-checking between moderator and note-taker, discussion on notes, and comparing them with audio recordings. Both the researchers thoroughly read the transcripts, to get familiarize with the data and to furnish the subsequent rounds of coding. Codes with similar concepts were grouped into explicit themes.

## Ethical consideration

The ethical permission to conduct this study was granted by the Advance Studies & Review Board of the Khyber Medical University, Pakistan (No. DIR/KMU-AS&RB/IE/000966 dated 17th January 2020). The names and exact ages of the interviewed participants were masked in this study to ensure their confidentiality. The secondary data obtained was de-identified to ensure that none of the patients' personal details are exposed.

## Results

### Quantitative analysis

A total of 7041 meals were observed including breakfast, lunch, and dinner of all the in-patients during the study period. Of all the meals with dietary errors, 64.71% were critical and 35.29% were non-critical in nature (Table 3). Majority of the total dietary errors occurred during the diet card preparation (52.94%), by the dietitians (70.59%), during weekdays (82.35%), at breakfast time (47.06%), and in the cardiac care unit (47.06%).

In Table 3, we observed that majority of the critical errors occurred during tray preparations (66.67%), meal provision (100%), diet card preparations (77.78%), by the dietitians (75.00%), service aides (100%), in all the meal types (~66%), among renal transplant patients (66.67%) and patients admitted in the cardiac care unit (87.50%). Among all the variables, only ward was found to be statistically significant.

**Qualitative analysis.**   To find out the reasons of the dietary errors, we interviewed a total of 11 staff members responsible for dietary errors. The saturation in responses was observed in 10 interviews and was confirmed in one more interview. Some of the staff in the food department attributed the dietary errors to the increased workload while others could not concentrate on work due to various reasons. The themes that emerged from the analysis of the interviews are as follow:

**Theme 1: Lack of back-up/accessory staff.**   The participants stated that hospital had limited number of employees in the food department and did not have a backup staff to replace any staff member on leave. One of the employees attributed the reason of dietary errors to the inability to handle increased demand load due to increased admissions. The food staff responsible for the diet error said;

*"The patient load was too high for me to handle single-handedly"*. (A male in his 20's)

**Table 3. The percentage wise distribution of dietary errors (N = 7041).**

| Variables | | Critical Errors N (%) | Non-Critical Errors N (%) | Total Errors N (%) | p-value* |
|---|---|---|---|---|---|
| **Overall** | | 11 (64.7) | 6 (35.3) | 17 (0.24) | |
| **Point of Error** | Diet Card Preparation | 7 (77.8) | 2 (22.2) | 9 (52.9) | 0.420 |
| | Diet Entry | 1 (33.3) | 2 (66.7) | 3 (17.7) | |
| | Packing | 0 (0) | 1 (100) | 1 (5.9) | |
| | Tray line | 2 (66.7) | 1 (33.3) | 3 (17.7) | |
| | Meal Provision | 1 (100) | 0 (0) | 1 (5.9) | |
| **Responsible Staff** | Dietitian | 9 (75.0) | 3 (25.0) | 12 (70.6) | 0.245 |
| | Unit Representative | 1 (33.3) | 2 (66.7) | 3 (17.6) | |
| | Cook | 0 (0) | 1 (100) | 1 (5.9) | |
| | Service Aide | 1 (100) | 0 (0) | 1 (5.9) | |
| **Weekday** | Yes | 10 (71.4) | 4 (28.6) | 14 (82.3) | 0.515 |
| | No | 1 (33.3) | 2 (66.7) | 3 (17.7) | |
| **Meal Type** | Breakfast | 5 (62.5) | 3 (37.5) | 8 (47.0) | 1.00 |
| | Lunch | 4 (66.7) | 2 (33.3) | 6 (35.3) | |
| | Dinner | 2 (66.7) | 1 (33.3) | 3 (17.7) | |
| **Ward** | Renal Transplant | 2 (66.7) | 1 (33.3) | 3 (17.7) | 0.037 |
| | Cardiac Care | 7 (87.5) | 1 (12.5) | 8 (47.1) | |
| | Gastroenterology | 1 (50.0) | 1 (50.0) | 2 (11.8) | |
| | Orthopedics | 0 (0) | 3 (100) | 3 (17.7) | |
| | Surgical | 0 (0) | 1 (100) | 1 (5.9) | |

*Fisher exact test.

Similarly, a food staff came back after a long leave and could not handle patient load on the first few days;

*"I was feeling tired as the patient load was too much. I could not concentrate after a long break".* (A male in his 20's)

The lack of concentration, in some instances, was attributed to increased workload by the participants especially in cases where rare situations were encountered which resulted in critical errors. Rare situations were reported as food allergies of some patients or presence of two patients with exact same names but different rooms.

*"I didn't notice the allergy note as it was not prominent. There are so many patients' diet to handle, so it went unnoticed".* (A female in her 20's)

*"The patient name was same and I didn't pay attention to different room numbers. That day there were lot of food trays for delivery to patients. I should have checked the names and room numbers before serving anything to patients".* (A male in his 40's)

According to participants, burnout due to working overtime to compensate for absent staff led to increased chances of errors. One of the food staff mentioned that;

*"Sometimes we need to perform extra shift to cover for the staff on leave, this becomes burden and tiresome as the working hours become too lengthy".* (A male in his 20's)

Another explicitly added;

*"The management does not have a backup staff to cover for staff on leave that may get absent for any reason including getting sick".* (A male in his 30's)

**Theme 2: Employees' personal and domestic issues.**   Another theme that emerged was the employees' personal and domestic issues because of which they were unable to concentrate at work. Some of the employees attributed the reasons for dietary error to distributed concentration at work as a result of incidents at home. For instance, one of the staff mentioned;

*"I was stressed as my spouse got injured in a road traffic accident, I needed a leave but could not get it, and therefore, I could not focus at work".* (A female in her 20's)

Another expressed a job related pressure from home,

*"My spouse wants me to leave the job and be at home".* (A female in her 20's)

Another participant had an exchange of words at home as a result of which the staff was depressed;

*"I had an argument with mother-in-law just few minutes before leaving home for duty. This made me depressed and I could not pay attention at work".* (A female in her 20's)

Similarly, one of the food staff mentioned being unable to get leave due to illness as a reason for dietary errors;

*"I didn't sleep well just because I was not feeling well, so I was disturbed during my duty or you can say I was absent minded".* (A female in her 20's)

## Discussion

Nutrition therapy is often used in conjunction with the medical treatment to support and optimize the overall treatment outcome of the hospitalized patients [17]. The delivery of correct therapeutic diet is of paramount importance, in this regard. The prevalence of critical dietary errors accounted for majority of the dietary errors in our study. This result is consistent with other observational studies where the proportion of critical errors was the highest [6,8]. The majority of dietary errors took place during the diet card preparation followed by diet entry into the computerized system and the tray line. These findings are comparable to a study conducted in Thailand, which also related majority of the dietary errors to diet card preparation in the diet flow process [18]. The diet entry and diet card preparation processes are crucial being the initial steps in the diet flow process. It is imperative that these steps remain error free to prevent the error carried all the way to the patient. The main reasons of dietary errors were the lack of back up staff and the employees' personal and domestic issues in this study.

The dietitians' had two tasks at hand: to prepare diet cards and to check the diets with prepared diet cards during the tray preparation. In this study, about 70% of errors were attributed to the dietitians. This result is congruent to the Australian study where dietitians made up for majority of dietary errors [6]. Manual diet card preparation is not only cumbersome but also increases chance of error in case of increased workload. The human error can be prevented by

getting the diet cards printed via computer systems or through automation as soon as the unit representative of the ward inputs the diet information into the computerized system [19]. Previous studies reported a significant reduction in dietary errors after introduction of automated diet cards' preparation in hospital settings [6,8,20]. To have error free diet cards, it is imperative to have error free diet entry into the computerized system by the unit representative. The error free diet entry in the system could be ensured by double entry computerized systems as evident from research [21]. Alternatively, a tablet or phone based e-application could be developed, with a digital checklist diet form, linked to the patient's electronic record system to prevent dietary errors [19]. Any diet entry that conflicts with the medical condition of the patient would generate an error upon incorrect diet selection. This system may have upfront cost but in the long run, would save expenses related to diet replacements, time, cooks' labor, and medical expenses in case of inaccurate diet consumption. It is noteworthy that none of the dietary errors were attributed to the floor staff. This may indicate that a visual check of the diet with patient notes is a fruitful process of detecting dietary errors.

The reduction of in-patient dietary errors is of highest priority to improve the quality of patient care and treatment outcome. Humans are emotional beings [22] and any factor associated with their emotions may affect their work performance [23]. The common reasons for the errors were related to human capacity and human needs, e.g. increased workload, personal and domestic issues, limited capacity to manage patient load, and overwork burn out. There was less probability that someone would be available at home over the weekdays to take care of emergency issues, if one arises like narrated by the participants in this study. Similarly, start of the day may not be good if the employee's concentration is not over the designated tasks [24,25]. It has been documented that occupational stress contributes to organizational inefficiencies especially in healthcare organizations where being in the right mind matters the most [26]. This warrants a need of stress management strategies and policies to be introduced in hospital for the staff to be fully available and efficient.

The private for-profit hospitals in low middle-income countries, like Pakistan, work on limited resources to maximize profits. In such countries, technology is expensive as compared to human resource. The findings of this study suggest that human errors could be minimized by adapting organizational behavior skills. For example, integrating empathy with the human resource management to improve employee job satisfaction. To do this, there should be a manageable workload on the employee. On-job training should be provided for capacity development of the food staff, and availability of back-up staff at all times to cater for increased patient load or replacement for staff on leave, so as not to compromise on quality and safety to in-patients.

## Limitations

The findings of this study come with a limitation on generalizability as the study was conducted in one hospital. However, this was the only hospital in the country that had implemented international standards and provided therapeutic dietary meals to in-patients as a part of treatment regimen. The study derives its strength from the large number of observations. Additional strength is the qualitative component of the study which explored actual causes of dietary errors in the meal flow process. This is a firm strength of the study as qualitative components have not been explored in any of the earlier studies. Another limitation of the study was the use of the questionnaire to guide the student researcher on the collection of secondary data from the hospital records on a daily basis. The purpose was to prevent the student researcher from deviation from collecting data not related to the research objectives. This prevented us to apply Cronbach's alpha to test the questionnaire's reliability as neither did it contain any latent constructs or likert scale, nor did we carry out a field survey.

## Recommendations

Based on the findings of this study, it is suggested that human errors could be minimized by adapting organizational behavior skills. For example, integrating empathy with the human resource management to improve employee job satisfaction. To do this, there should be a manageable workload on the employee. On-job training should be provided for capacity development of the food staff, and availability of back-up staff at all times to cater for increased patient load or replacement for staff on leave, so as not to compromise on quality and safety to in-patients.

## Conclusion

In this study, critical dietary errors made up the majority of dietary errors in the hospital food department. Human factors were found to be the primary causes of these errors. It is proposed that the hospital may ensure and adhere to the principles of organizational behavior for contented employees that may result in error-free therapeutic food services to the patients. This may enhance patient safety, patient satisfaction, and patient well-being for better clinical outcomes.

## Supporting information

**S1 Questionnaire.**
(DOCX)

## Author Contributions

**Conceptualization:** Amanullah Khan, Sidra Malik, Fayaz Ahmad, Naveed Sadiq.

**Data curation:** Amanullah Khan, Sidra Malik, Fayaz Ahmad, Naveed Sadiq.

**Formal analysis:** Amanullah Khan, Fayaz Ahmad, Naveed Sadiq.

**Investigation:** Amanullah Khan.

**Methodology:** Amanullah Khan, Fayaz Ahmad, Naveed Sadiq.

**Project administration:** Amanullah Khan.

**Software:** Amanullah Khan, Naveed Sadiq.

**Writing – original draft:** Amanullah Khan, Sidra Malik, Fayaz Ahmad, Naveed Sadiq.

**Writing – review & editing:** Amanullah Khan, Sidra Malik, Fayaz Ahmad, Naveed Sadiq.

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
