## [Decision Letter · Decision Letter 0]

11 Apr 2022

PONE-D-21-39046The importance of human factors in therapeutic dietary errors of a hospital: A mixed-methods studyPLOS ONE

Dear Dr. Sadiq,

Thank you for submitting your manuscript to PLOS ONE. After careful consideration, we feel that it has merit but does not fully meet PLOS ONE’s publication criteria as it currently stands. Therefore, we invite you to submit a revised version of the manuscript that addresses the points raised during the review process.

We look forward to receiving your revised manuscript.

Kind regards,

Yong-Hong Kuo

Academic Editor

PLOS ONE

Journal Requirements:

Additional Editor Comments:

Based on the reviewers' recommendations and comments, I recommend Major Revision. Please address their concerns in the revision.

Reviewers' comments:

Reviewer's Responses to Questions

**Comments to the Author**

1. Is the manuscript technically sound, and do the data support the conclusions?

Reviewer #1: Yes

Reviewer #2: No

Reviewer #3: Yes

Reviewer #4: No

2. Has the statistical analysis been performed appropriately and rigorously? 

Reviewer #1: Yes

Reviewer #2: No

Reviewer #3: Yes

Reviewer #4: No

3. Have the authors made all data underlying the findings in their manuscript fully available?

Reviewer #1: No

Reviewer #2: No

Reviewer #3: Yes

Reviewer #4: No

4. Is the manuscript presented in an intelligible fashion and written in standard English?

Reviewer #1: Yes

Reviewer #2: Yes

Reviewer #3: Yes

Reviewer #4: No

5. Review Comments to the Author

Reviewer #1: Check the grammar in the following sentence:

106 1) Diet entry

107 Diet entry was the first point where patient’s diet was entered the system by the Unit

108 Representative (UR) and was the primary source of information provided to the dietary

109 staf

In the Abstract it is stated that you employed SPSS v 20.

165 All the quantitative variables were categorical in nature. The row percentages were calculated

166 for all the independent variables against the outcome variable. We calculated frequencies and

167 percentages for all the critical and non-critical errors using SPSS version 25 (13).

Which version was rightly used.

Reviewer #2: There is no comment. I think both your method and your discussion have technical problem. Also your manuscript is not fit to this journal. I believe it would be great if you submit it to a journal which completely fit to your study.

Reviewer #3: In general, the article has been edited correctly, but some parts of the article need to be changed

Due to the non-native language of the article writing, my judgment about writing English may not be correct, but regarding other parts of the article, I expressed my views scientifically as a PhD student.

Reviewer #4: PONE-D-21-39046

Re: “The importance of human factors in therapeutic dietary errors of a hospital: A mixed methods study”

Thank you for asking me to review this manuscript. I have the following Comments:

Abstract: It should be rewritten in a structure format as per PLOS journal guidelines.

Introduction:

The authors quoted 11 references under introduction with only three short paragraphs. The introduction will benefit from details explanation related to the objectives of this study. Start with the definition of dietary errors and its examples.

Consider replacing “err” on line 71 by “error”.

Methods:

This section requires major revision.

The study tool used for this study on line 87-88, should be referenced or accessed through a link.

The validity of questionnaire and their Cronbach’s alpha and reliability scores should be reported after data collection.

Explain in more details’ inclusion and exclusion criteria of the study.

How were the data extracted? What were the sampling technique? What was the sampling equation to calculate sample size? How did the authors come up with 7041?

Explain in detail the independents variables?

Qualitative study:

The one conducting interview should have experience in qualitative research.

Experts in qualitative research should supervised data collection and hold feedback sessions shortly after the interview.

All interviews should be recorded and transcribed in full to verify the accuracy of responses.

Two members of the research team independently should analyze the transcribed responses and read them on multiple times to familiarize self with the contents and categorize it in a meaningful way.

All these points should be explained and incorporated under the design of qualitative study.

Quantitative study:

It should be explained in details and the collected data should be reported. Authors should do statistical analysis for quantitative data.

The authors stated on lines 144-145 that “The dietary information from the comatose patients, and patients on nasogastric feed were excluded from the study” Why? No patients interview was carried out in this study.

What was the definition of dietary errors in this study?

Table 1 and 2 should be moved under results.

What was ADA in Table 2 stand for?

Ethical consideration: needs to be under subheading after methods. There were issues with confidentiality and power imbalance with the recruitment process and I was unsure what the recruitment process was?

Results: Should be rewritten once the methods revised.

Statistical analysis should be done and report p-values. Furthermore, relationship between dependent and independent variables should be done from statistical analysis point of view.

Explain in details the personal and familial issues of the participants in relation to dietary errors.

Conclusion: It should be re-written with a major focus on the main results of the study. In addition, one statement of recommendation based on the findings of the study should be mentioned under conclusion. The statement about deaths in this study (lines 315-316), should be deleted as this was not investigated in this study.

Limitations of the study: should explain in details and logical ways. Personal, recall, misinterpretation biases and Hawthorne effect were possibilities in this study

References: The manuscript presents several references which were outdated, and some were irrelevant. The references should be revised, updated, and only used if it were relevant to the study. It also should follow the guidelines of the journal.

6. PLOS authors have the option to publish the peer review history of their article (what does this mean?). If published, this will include your full peer review and any attached files.

Reviewer #1: No

Reviewer #2: No

Reviewer #3: **Yes: **Samira Raoofi (PhD student)

Reviewer #4: No

---

## [Author Response · Author response to Decision Letter 0]

21 May 2022

May 21, 2022

Yong-Hong Kuo

Subject: Response to Reviewers’ Comments

Dear Hon. Yong-Hong Kuo,

We thank the reviewers for taking out precious time to provide valuable comments for improving the manuscript. The comment serves to shape the manuscript to its perfection. Thank you for the constructive feedback on our manuscript, which we have carefully incorporated in the revised manuscript. 

Please find responses to each comment below.

Thank you very much.

Response to Reviewer # 1:

Comment 1: Reviewer #1: Check the grammar in the following sentence:

106 1) Diet entry

107 Diet entry was the first point where patient’s diet was entered the system by the Unit

108 Representative (UR) and was the primary source of information provided to the dietary

109 staf

Response: We apologize for the inconvenience. The grammatical error has been rectified as per the reviewer’s suggestion. Thank you very much for noticing it.

Comment 2: In the Abstract it is stated that you employed SPSS v 20.

165 All the quantitative variables were categorical in nature. The row percentages were calculated

166 for all the independent variables against the outcome variable. We calculated frequencies and

167 percentages for all the critical and non-critical errors using SPSS version 25 (13).

Which version was rightly used?

Response: Thank you for noticing it. This happened as a result of the update of the software from thesis writing to the manuscript. This has been corrected now at both the places.

Response to Reviewer # 2:

Comment: There is no comment. I think both your method and your discussion have technical problem. Also your manuscript is not fit to this journal. I believe it would be great if you submit it to a journal which completely fit to your study.

Response: Thank you very much for taking time to review our manuscript. We looked at the journal information and scope of the journal before submitting the manuscript. In our view, our manuscript is pertinent and aligns with the aims and scope of the Plos One journal. Thank you once again.

Response to Reviewer # 3:

In general, the article has been edited correctly, but some parts of the article need to be changed.

Due to the non-native language of the article writing, my judgment about writing English may not be correct, but regarding other parts of the article, I expressed my views scientifically as a PhD student.

Response: We thank you for taking out the time to review our manuscript in spite that English is not your native language. Your review regarding our manuscript as a PhD student is much appreciated. 

We believe that you are the reviewer who provided comments in the PDF document. In the light of this assumption, please find responses to your comments below.

Comment 1: It is better to compile a more concise abstract.

Response: The abstract followed journal guidelines and was tacitly limited to 249 words. On your recommendation, we have modified it and tried our best to keep it succinctly concise to give a brief flavor of all done in a nut shell. Thank you.

Comment 2: Table findings are not clear to contacts.

Response: The tables 1 and 2 are not results, rather these are the classification tables provided by the hospital to classify the dietary errors. Table 1 is about those dietary errors that could be noticed with the naked eye. Table 2 is about those errors that could not be noticed with a naked eye. However, patient’s development of symptoms later, as a result of consumption of inaccurate diet, is verified through a biochemical analysis as laid out in Table 2 against each advised diet to determine the inaccuracy in diet provision. The text associated with the tables have been rephrased to remove ambiguity. Thank you.

Comment 3: Regarding quantitative studies, the steps of the study should be mentioned exactly as mentioned in the qualitative studies.

Response: Thank you for your great comment. We have now accumulated all the steps of Quantitative methods under the heading of “Quantitative Study”. Similarly we have accumulated all the scattered information on qualitative methods under the heading of “Qualitative Study”.

Comment 4: Regarding the qualitative method, it is better to explain it as follows:

Qualitative study

How is the interview guide developed?

What resources have been used to compile the guide?

Response:

The topic guide was developed by the researchers keeping in view the objective of study (searching for reasons of errors), with further feedback from a qualitative research expert. As this was a much focused objective, we didn’t use any other resources to develop the topic guide. This is now explained on page 9, line #195-199. Thank you.

The information of the research community should be mentioned accurately.

Response:

Your point is well taken. We have now mentioned (page 9 line# 202-207) the details of the researcher who conducted the interviews. Thank you.

What type of sampling was used in selecting the statistical population?

Response: 

We used purposive sampling technique, the most frequently used approach in qualitative research, mentioned on page 9, line #191-192. Thank you.

How were the interviews conducted?

Response:

This is mentioned on page 9, line #191-207. Thank you.

The results should be categorized as main and secondary themes and final codes in a table.

Response:

Thank you for bringing this up. The qualitative results are now presented under two emerging themes (page 11, line #238 and page 12, line #275). We didn’t opt for a table of qualitative results for two reasons. (1) There are already three tables for quantitative section of the study and (2) the reasons for dietary errors are reasonably explicit under the two themes. Thank you.

What has been the reliability and validity of your qualitative study?

Response:

Among the aspects of Credibility (validity) and Dependability (reliability) of a qualitative study, some features are touched upon and can be seen in methods and results section. These included: debriefing sessions, use of probes, informed consent, use of representative quotes, overall clearly mentioned methods. The previously scattered information of qualitative section in methods has now been gathered under one title “qualitative study” on page 9. Thank you.

In the end, how was the data analysis and with what software?

Response:

We used deductive-inductive thematic analysis approach, mentioned on page 9, line #202 with due citation. Moreover, we did manual coding without using any software. Thank you.

Comment 5: It is better to establish correlations between the contents in the findings.

Response: All the data that was collected through the secondary sources is categorical in nature. The correlations, therefore, do not apply. Moreover, the correlations are not applicable as the objective was not find any statistical association. Thank you.

Comment 6: The results should be categorized according to the main themes and its descriptions.

And at the end, write the key sentences of the interviews.

Response: Thank you for pointing this out. The results are now organized to the main themes, its description, and the verbatims.

Comment 7: In any case, the discussion has been done correctly, but it is better to compare with more studies, and given that the study includes two categories of quantitative and qualitative results, the number of studies compared is not enough for discussion.

Response: The reviewer is right that the main findings of the study should be compared with further studies. However, it is unfortunate that we could find only two studies related to the quantitative findings of the study. The research literature regarding the therapeutic diets is extremely scarce, else we would have added further studies. 

However, studies related to the findings of the qualitative portion of the manuscript have been added to support the claims. Thank you.

Comment 8: The conclusion section should be revised and in this section the findings should not be repeated, but executive strategies should be developed to improve the current situation and suggestions for future research should be stated along with study limitations.

Response: The conclusion section has been revised a bit. The conclusion section reflects the synthesis of key points and way forward. The limitations are usually a part of the discussion section. The limitations could be found in the last paragraph of the discussion section. A separate section on recommendations is now added before conclusions. Thank you.

Response to Reviewer # 4:

Comment 1:

Abstract: It should be rewritten in a structure format as per PLOS journal guidelines. 

Response:

The abstract follows guidelines given by the PLOS One at:

Thank you.

Comment 2:

Introduction: 

The authors quoted 11 references under introduction with only three short paragraphs. The introduction will benefit from details explanation related to the objectives of this study. Start with the definition of dietary errors and its examples. 

Consider replacing “err” on line 71 by “error”.

Response:

There hasn’t been much research on therapeutic diets provided in the hospital systems. This is why there isn’t much material available on it. As per the reviewer’s suggestion, we have lengthen the introduction section, keeping in view that it should not deviate from the objectives of the study. The definition of dietary errors and its examples have also been provided now. 

“To err is human” is considered a standard phrase these days and used by WHO as such as well. Please follow the following link: https://www.who.int/news-room/fact-sheets/detail/patient-safety Heading “Why does patient harm occur?” first line of the second paragraph.

If the reviewer wants us to replace it with the term “error”, we would be still happy to do it. Thank you.

Comment 3:

Methods: 

This section requires major revision.

The study tool used for this study on line 87-88, should be referenced or accessed through a link.

Response:

We apologize for the inconvenience caused at this stage. However, the study tool cannot be hyperlinked at this stage, as per journal requirements. It will be accessed through a link when and if it gets published. In current form, it could be accessed at the very end of the manuscript where the page title states “Supporting Document 1: Study Tool” via link auto-generated (at the end of the document) by the journal during the submission stage. Within the manuscript, it has been referenced to “supporting document 1 title Study Tool” on page 5 line # 105. Thank you.

Comment 4:

The validity of questionnaire and their Cronbach’s alpha and reliability scores should be reported after data collection.

Response:

The quantitative portion of the study collected information from secondary data, extracted from the hospital records. The questionnaire was not meant to conduct a survey in the field. The statements have been amended in the manuscript to reflect better on the process of quantitative data collection. The validity and the associated measures, therefore, do not apply for both the quantitative and qualitative components of the study. Thank you.

Comment 5:

Explain in more details’ inclusion and exclusion criteria of the study. 

Response:

All the diets provided within the hospital system in one month were included in the study. Only diets to those on nasogastric tube and comatose patients were excluded as these diets used to come from the hospital’s pharmacy and not the hospital’s kitchen, e.g. glucerna, and such diets were not a part of the hospital’s meal flow process. Both the inclusion and exclusion criteria have been expanded as per the reviewer’s comment. Thank you.

Comment 6:

How were the data extracted? What were the sampling technique? What was the sampling equation to calculate sample size? How did the authors come up with 7041?

Response:

The secondary quantitative data were extracted from the hospital records on a daily basis. When an error was encountered, it was notified to the hospital’s computer system, the source of error was traced and the relevant personnel was/were interviewed. All the diets within one month of the study period were taken into account. All these diets, three per day per patient, accounted to 7041. This is why we deployed census sampling technique, also mentioned in the text on page 8 line 160. Thank you.

Comment 7:

Explain in detail the independents variables?

Response:

The details on the independent variables are available on page 8 lines 177-184. Thank you.

Comment 8:

Qualitative study: 

The one conducting interview should have experience in qualitative research. 

Response: 

The interviews were conducted by a student of Masters in Public Health, who was trained in qualitative research, and this is now mentioned on page 9, line 199-201. Thank you.

Comment 9:

Experts in qualitative research should supervised data collection and hold feedback sessions shortly after the interview.

All interviews should be recorded and transcribed in full to verify the accuracy of responses.

Two members of the research team independently should analyze the transcribed responses and read them on multiple times to familiarize self with the contents and categorize it in a meaningful way. 

All these points should be explained and incorporated under the design of qualitative study. 

Response:

Almost all of these comments were already there but probably due to scattered information of quantitative and qualitative sections in methods, this information didn’t seem explicit. We have now gathered all the qualitative study methods under one title “qualitative study” on page 9, satisfying all your comments. Thank you very much for your constructive comment.

Comment 10:

Quantitative study: 

It should be explained in details and the collected data should be reported. Authors should do statistical analysis for quantitative data.

Response:

The statistical analyses were performed according to the objectives of the study. The percentages have been reported that best suit the objectives of this manuscript. Thank you.

Comment 11:

The authors stated on lines 144-145 that “The dietary information from the comatose patients, and patients on nasogastric feed were excluded from the study” Why? No patients interview was carried out in this study.

Response:

All the diets provided within the hospital system in one month were included in the study. Only diets to those on nasogastric tube and comatosed patients were excluded as these diets used to come from the hospital’s pharmacy and not the hospital’s kitchen, e.g. glucerna, and such diets were not a part of the hospital’s meal flow process. This explanation has now been added in the manuscript. Thank you.

Comment 12:

What was the definition of dietary errors in this study?

Response:

Thank you for pointing this out. The dietary errors were not explicitly mentioned in the introduction section. The tables 1 and 2 denote the dietary errors: the first column of each table denoting the diet that was recommended and the second column denoting the diet that was received in error. Now the introduction section contains the definition of dietary errors and its types. Please consult page 3 lines 64-65. Thank you.

Comment 13:

Table 1 and 2 should be moved under results. 

Response:

The placement of tables follow the journal guidelines, i.e., “Tables should be included directly after the paragraph in which they are first cited.” (https://journals.plos.org/plosone/s/file?id=wjVg/PLOSOne_formatting_sample_main_body.pdf)

Tables 1 and 2 are not the results but contain the detailed information on the dietary errors and its types. Thank you.

Comment 14:

What was ADA in Table 2 stand for?

Response:

Thank you for noticing it. ADA refers to the American Diabetes Association. This indicated the diabetic diet recommended by the American Diabetes Association for the diabetic patients. This acronym has been detailed in Table 2 now. Thank you.

Comment 15:

Ethical consideration: needs to be under subheading after methods. There were issues with confidentiality and power imbalance with the recruitment process and I was unsure what the recruitment process was?

Response:

The ethical consideration has been moved as a subheading after methods. The confidentiality statements have been added for both quantitative and qualitative components under the ethical consideration subheading. 

The quantitative study was observational in nature and all the data was extracted on a daily basis over one month, therefore, the concept of statistical power does not apply here. Census sampling was used to collect all the data on a daily basis for one month. For the qualitative interviews, the sampling was purposive as only those personnel were interviewed who were reported for dietary error. Thank you.

Comment 16:

Results: Should be rewritten once the methods revised.

Response:

Subheadings have been added in the manuscript for clarity purposes. The results section has been modified to reflect further clarity. Thank you.

Comment 17:

Statistical analysis should be done and report p-values. Furthermore, relationship between dependent and independent variables should be done from statistical analysis point of view. 

Response:

The most pertinent statistical analyses are the percentages in this study. As the objective of the study was neither to explore the associations nor the factors associated with the outcome variable nor to test a hypothesis, therefore, inferential statistics were not deployed, rather only descriptive statistics were reported. Another reason for reporting only percentages was to assess the prevalence of dietary errors. Losing one precious life due to dietary error is as serious as losing multiple lives. This is why the percentages mattered the most, in our opinion, here. Thank you.

Comment 18:

Explain in details the personal and familial issues of the participants in relation to dietary errors.

Response: 

This is described in the results and discussion sections both. The results section has verbatims in the italicized text to give an idea on what was going on in the personal lives of the interviewed participants that effected their work at the hospital. This has been referenced with theories/published research papers. Thank you.

Comment 19:

Conclusion: It should be re-written with a major focus on the main results of the study. In addition, one statement of recommendation based on the findings of the study should be mentioned under conclusion. The statement about deaths in this study (lines 315-316), should be deleted as this was not investigated in this study. 

Response:

The conclusion has been modified as per the reviewer’s suggestion. Thank you.

Comment 20:

Limitations of the study: should explain in details and logical ways. Personal, recall, misinterpretation biases and Hawthorne effect were possibilities in this study

Response:

Personal biases could have been there but this is a universal problem to all the qualitative studies. However, the personal bias was minimal in this study as the researchers verified some of the claims of the study through the hospital records, e.g. the leave application was submitted but not granted to one/some of the participants.

Recall bias was also minimized as the researchers interviewed the participants on the same day of the error notified. 

Misinterpretation bias was also minimized. The jargon was not used, rather the interviews utilized local language of the study tool.

There could have been Hawthorne Effect if the study participants were observed directly. However, none of the participants at each level of the food preparation were observed directly. Only when a dietary error was reported, the respective participant was contacted with their permission to explore the causes of error.

Thank you for asking for explanation on these.

Comment 21:

References: The manuscript presents several references which were outdated, and some were irrelevant. The references should be revised, updated, and only used if it were relevant to the study. It also should follow the guidelines of the journal.

Response:

The research topic has not been studied much. The references found in the subject area are from year 2001 and onwards and these are the only references that are available to this research topic. The references have been updated (where applicable). Kindly note that the references have added to justify/support the statements/claims made in the study, these are, therefore, need to be cited. Thank you.

---

## [Decision Letter · Decision Letter 1]

30 May 2022

PONE-D-21-39046R1The importance of human factors in therapeutic dietary errors of a hospital: A mixed-methods studyPLOS ONE

Dear Dr. Sadiq,

Thank you for submitting your manuscript to PLOS ONE. After careful consideration, we feel that it has merit but does not fully meet PLOS ONE’s publication criteria as it currently stands. Therefore, we invite you to submit a revised version of the manuscript that addresses the points raised during the review process.

We look forward to receiving your revised manuscript.

Kind regards,

Yong-Hong Kuo

Academic Editor

PLOS ONE

Journal Requirements:

Additional Editor Comments (if provided):

Four of the reviewers from the last round were invited to review this revision. Three of them accepted the invitation and returned the reports. These three reviewers recommend Accept, Minor Revision, and Reject.

Based on their recommendations and comments, I suggest minor revision. Please address the comments from the reviewers for possible publication at PLOS ONE.

Reviewers' comments:

Reviewer's Responses to Questions

**Comments to the Author**

1. If the authors have adequately addressed your comments raised in a previous round of review and you feel that this manuscript is now acceptable for publication, you may indicate that here to bypass the “Comments to the Author” section, enter your conflict of interest statement in the “Confidential to Editor” section, and submit your "Accept" recommendation.

Reviewer #2: (No Response)

Reviewer #3: All comments have been addressed

Reviewer #4: (No Response)

2. Is the manuscript technically sound, and do the data support the conclusions?

Reviewer #2: Partly

Reviewer #3: Yes

Reviewer #4: Yes

3. Has the statistical analysis been performed appropriately and rigorously? 

Reviewer #2: No

Reviewer #3: Yes

Reviewer #4: No

4. Have the authors made all data underlying the findings in their manuscript fully available?

Reviewer #2: (No Response)

Reviewer #3: Yes

Reviewer #4: Yes

5. Is the manuscript presented in an intelligible fashion and written in standard English?

Reviewer #2: No

Reviewer #3: Yes

Reviewer #4: Yes

6. Review Comments to the Author

Reviewer #2: (No Response)

Reviewer #3: Thanks to the dear authors for editing the article, for

All the comments related to me have been answered logically and there is no obstacle for me to accept them.

Reviewer #4: Thank you very much for allowing me to review this manuscript for the second time>

Most of the required corrections have been made and the manuscript had been improved scientifically. The authors responded to majority of my comments and provided valuable information which are important for the readers. I further have the followings minor comments:

1. The validity of questionnaire and their Cronbach’s alpha and reliability scores should be reported after data collection. This was communicated to the authors, however they felt that this do not apply for both quantitative and qualitative components of the study as the questionnaire was not meant to conduct a survey in the field. Regardless, as long as the questionnaire was used in this study, its validity and their Cronbach’s alpha and reliability scores should be calculated.

2. Authors should do statistical analysis for quantitative data. Furthermore, relationship between dependent and independent variables should be done from statistical analysis point of view. The authors responded to this issue as the objective of the study was neither to explore the associations nor the factors associated with the outcome variable nor to test a hypothesis, therefore, inferential statistics were not deployed, rather only descriptive statistics were reported. I do not think this is the case as further statistical analysis will help to achieve the objectives of the study and use independent variables in this study to predict the value of dependent variable (dietary errors). I think such statistical analysis will strengthen the findings of this study.

7. PLOS authors have the option to publish the peer review history of their article (what does this mean?). If published, this will include your full peer review and any attached files.

Reviewer #2: No

Reviewer #3: **Yes: **Samira Raoofi: Ph.D. student in Health Care Management

Iran University of Medical Sciences.

Reviewer #4: No

---

## [Author Response · Author response to Decision Letter 1]

5 Jul 2022

July 05, 2022

Yong-Hong Kuo

Subject: Response to Reviewers’ Comments

Honorable Yong-Hong Kuo,

We thank the reviewers for taking out precious time to review our manuscript. The comments serve to shape the manuscript to its perfection. Thank you for the constructive feedback on our manuscript, which we have carefully incorporated in the revised manuscript.

Please find responses to each comment below.

Thank you very much.

Response to Journal Requirements:

Response: To our knowledge, none of the papers cited have been retracted. We double checked all the articles for retraction. However, the reference number 4 had an old link that was changed after the submission of the manuscript by the publishers. The link has been updated now for the paper referenced as number 4. Thank you for noticing this.

P.S: The referenced article 4 does not depict tracked changes in the “references” section for some reasons.

Response to Reviewer # 4:

Thank you very much for allowing me to review this manuscript for the second time.

Most of the required corrections have been made and the manuscript had been improved scientifically. The authors responded to majority of my comments and provided valuable information which are important for the readers. I further have the followings minor comments:

1. The validity of questionnaire and their Cronbach’s alpha and reliability scores should be reported after data collection. This was communicated to the authors, however they felt that this do not apply for both quantitative and qualitative components of the study as the questionnaire was not meant to conduct a survey in the field. Regardless, as long as the questionnaire was used in this study, its validity and their Cronbach’s alpha and reliability scores should be calculated.

Response: The questionnaire was formed to assist the student researcher to extract data from secondary sources (hospital records) so that only that information is gathered that is related to the research question and to prevent any deviation or distraction for the student researcher. It should be noted that neither latent variables were constructed nor any field survey was conducted in our study. Also, the questionnaire did not use any Likert scale. The honorable reviewer remarks that as the word "questionnaire" has been used, therefore, Cronbach's alpha should be applied regardless. We tried to get more information on this topic by looking at the literature and realized this doesn't apply to our research. Please follow the articles below that justify our claim:

1. Making sense of Cronbach's alpha 

https://www.ncbi.nlm.nih.gov/pmc/articles/PMC4205511/

2. The Use of Cronbach’s Alpha When Developing and Reporting Research Instruments in Science Education 

https://link.springer.com/article/10.1007/s11165-016-9602-2

3. Cronbach Alpha Coefficient 

https://www.sciencedirect.com/topics/nursing-and-health-professions/cronbach-alpha-coefficient

4. Using and Interpreting Cronbach’s Alpha 

https://data.library.virginia.edu/using-and-interpreting-cronbachs-alpha/

5. Cronbach’s Alpha: Definition, Interpretation, SPSS 

https://www.statisticshowto.com/probability-and-statistics/statistics-definitions/cronbachs-alpha-spss/

6. Cronbach’s Alpha and Semantic Overlap Between Items: A Proposed Correction and Tests of Significance 

https://www.ncbi.nlm.nih.gov/pmc/articles/PMC8867700/

If the concern is over the word “questionnaire”, we can rephrase the manuscript that the data were drawn from secondary sources as an alternative, skipping the term questionnaire at all. We wrote the manuscript in complete honesty. This is why we explained everything in detail in addition to allow for reproducibility of the research. However, to address the concern of the reviewer in this scenario, we have added the issue of Cronbach’s alpha in the limitations section of our manuscript, which could be found at page 17 lines 365-370.

2. Authors should do statistical analysis for quantitative data. Furthermore, relationship between dependent and independent variables should be done from statistical analysis point of view. The authors responded to this issue as the objective of the study was neither to explore the associations nor the factors associated with the outcome variable nor to test a hypothesis, therefore, inferential statistics were not deployed, rather only descriptive statistics were reported. I do not think this is the case as further statistical analysis will help to achieve the objectives of the study and use independent variables in this study to predict the value of dependent variable (dietary errors). I think such statistical analysis will strengthen the findings of this study.

Response: Majority of the times, inferential statistics are used when one is testing a hypothesis or wants to explore the associations. Our study did not have any such objectives. We used only descriptive statistics because a critical error in the hospital setting could hamper a patient's life, regardless if it's statistically significant or not. In our expertise, the use of inferential statistics doesn't justify in this manuscript as the factors associated with the errors have been explored in the qualitative component of the manuscript. 

However, in the light of making the manuscript a bit more scientific as per reviewer’s comment, we applied the inferential statistics as it would not jeopardize the study objectives. The Fisher-exact based p-values have been reported to the Table 3 on pages 10-11. A bit of the description has also been added to the text on page 9, lines 187-189 and page 11, lines 231-232. Thank you.

---

## [Decision Letter · Decision Letter 2]

15 Aug 2022

The importance of human factors in therapeutic dietary errors of a hospital: A mixed-methods study

PONE-D-21-39046R2

Dear Dr. Sadiq,

We’re pleased to inform you that your manuscript has been judged scientifically suitable for publication and will be formally accepted for publication once it meets all outstanding technical requirements.

Kind regards,

Yong-Hong Kuo

Academic Editor

PLOS ONE

Additional Editor Comments (optional):

All the referees' concerns have been addressed. I recommend Accept.

Reviewers' comments:

Reviewer's Responses to Questions

**Comments to the Author**

1. If the authors have adequately addressed your comments raised in a previous round of review and you feel that this manuscript is now acceptable for publication, you may indicate that here to bypass the “Comments to the Author” section, enter your conflict of interest statement in the “Confidential to Editor” section, and submit your "Accept" recommendation.

Reviewer #2: All comments have been addressed

2. Is the manuscript technically sound, and do the data support the conclusions?

Reviewer #2: Yes

3. Has the statistical analysis been performed appropriately and rigorously? 

Reviewer #2: Yes

4. Have the authors made all data underlying the findings in their manuscript fully available?

Reviewer #2: Yes

5. Is the manuscript presented in an intelligible fashion and written in standard English?

Reviewer #2: Yes

6. Review Comments to the Author

Reviewer #2: Thank you for your revision, it has been perfectly done.

7. PLOS authors have the option to publish the peer review history of their article (what does this mean?). If published, this will include your full peer review and any attached files.

Reviewer #2: No

---

## [Editor Report · Acceptance letter]

17 Aug 2022

PONE-D-21-39046R2 

The importance of human factors in therapeutic dietary errors of a hospital: A mixed-methods study 

Dear Dr. Sadiq:

I'm pleased to inform you that your manuscript has been deemed suitable for publication in PLOS ONE. Congratulations! Your manuscript is now with our production department. 

Kind regards, 

on behalf of

Dr. Yong-Hong Kuo 

Academic Editor

PLOS ONE